# *In-Vitro* MPI-guided IVOCT catheter tracking in real time for motion artifact compensation

**Florian Griese** [1,2☯] *, **Sarah Latus**[3☯], **Matthias Schlüter** [3], **Matthias Graeser**[1,2], **Matthias Lutz**[4], **Alexander Schlaefer**[3], **Tobias Knopp**[1,2]

**1** Institute for Biomedical Imaging, Hamburg University of Technology, Hamburg, Germany, **2** Section for Biomedical Imaging, University Medical Center Hamburg-Eppendorf, Hamburg, Germany, **3** Institute of Medical Technology, Hamburg University of Technology, Hamburg, Germany, **4** Department of Internal Medicine, University Medical Center Schleswig-Holstein, Kiel, Germany

☯ These authors contributed equally to this work.
* florian.griese@tuhh.de

**Data Availability Statement:** The raw data of the IVOCT and MPI measurements are uploaded at https://doi.org/10.5281/zenodo.3554935.

## Abstract

### Purpose

Using 4D magnetic particle imaging (MPI), intravascular optical coherence tomography (IVOCT) catheters are tracked in real time in order to compensate for image artifacts related to relative motion. Our approach demonstrates the feasibility for bimodal IVOCT and MPI *in-vitro* experiments.

### Material and methods

During IVOCT imaging of a stenosis phantom the catheter is tracked using MPI. A 4D trajectory of the catheter tip is determined from the MPI data using center of mass sub-voxel strategies. A custom built IVOCT imaging adapter is used to perform different catheter motion profiles: no motion artifacts, motion artifacts due to catheter bending, and heart beat motion artifacts. Two IVOCT volume reconstruction methods are compared qualitatively and quantitatively using the DICE metric and the known stenosis length.

### Results

The MPI-tracked trajectory of the IVOCT catheter is validated in multiple repeated measurements calculating the absolute mean error and standard deviation. Both volume reconstruction methods are compared and analyzed whether they are capable of compensating the motion artifacts. The novel approach of MPI-guided catheter tracking corrects motion artifacts leading to a DICE coefficient with a minimum of 86% in comparison to 58% for a standard reconstruction approach.

### Conclusions

IVOCT catheter tracking with MPI in real time is an auspicious method for radiation free MPI-guided IVOCT interventions. The combination of MPI and IVOCT can help to reduce motion artifacts due to catheter bending and heart beat for optimized IVOCT volume reconstructions.

**Funding:** F.G., M.G. and T.K. thankfully acknowledge the financial support by the German Research Foundation (DFG, grant number KN 1108/2-1) and the Federal Ministry of Education and Research (BMBF, grant number 05M16GKA). A.S. acknowledges partial support by the German Research Foundation (DFG, grant number SCHL 1844/2-1/2). We acknowledge support for the Open Access fees by Hamburg University of Technology (TUHH) in the funding programme Open Access Publishing. The funders had no role in study design, data collection and analysis, decision to publish, or preparation of the manuscript.

# 1 Introduction

Optical coherence tomography (OCT) enables a high-resolution imaging of tissue structures [1–3]. In the field of cardiovascular diseases intravascular OCT (IVOCT) imaging is applied to assess the vascular wall structures and observe plaque formations and related stenosis lengths [4, 5]. IVOCT highly benefits from a second imaging modality in order to align its catheter tip position within the global coordinate system of the patient. Using digital subtraction angiography (DSA), ionizing radiation is introduced and only 2D projections of the catheter tip positions are observed. Different methods have been presented to determine the 3D vascular shape using a combination of IVOCT and angiographic images. For example, a co-registration of both imaging modalities is applied to align the images to each other [6–9]. An improved 3D volume reconstruction method uses the information of both the vessel center line as well as the 3D catheter trajectory determined in bi-plane angiographic frames [10]. Most of the recent volume reconstruction methods assume a static imaging scenario neglecting heart beat motion, arterial vasomotion, and catheter bending leading to motion artifacts. Nevertheless, several publications depict a relevant influence of motion artifacts on the IVOCT volume reconstructions. For example, an irregular formation of stent struts are related to heart beat motion [11, 12]. In a pre-clinical scenario a setup for ECG triggered IVOCT imaging with a duration of less than one second is proposed [13], hence heart beat motion artifacts can be minimized. Micro-motor catheters are proposed in order to deal with the problem of imaging artifacts due to bending of proximally rotated catheters [13, 14]. However, the miniaturization of high-speed motors is a challenging and expensive task. Thus, a medically approved IVOCT catheter with micro motor has not been presented yet. Consequently, motion artifacts due to catheter bending and arterial vasomotion still arise in clinical scenarios and have an influence on the quantification of plaque formations. In addition, a contrast agent (iodine) is necessary for DSA imaging, which can be problematic in some patients with kidney diseases [15–17].

As an alternative, magnetic particle imaging (MPI) spatially resolves the distribution of superparamagnetic iron oxide nanoparticles (SPION) in 4D at high temporal resolution by using the particle's non-linear magnetization characteristics [18, 19]. MPI applies static and oscillating magnetic fields to visualize the SPIONs. Thus in contrast to DSA, no ionizing radiation is induced to the patient. The changing magnetic fields are operated within the safety constraints of the peripheral nerve stimulation [20] and specific absorption rate (SAR) [21–23]. The SPIONs are biodegradable and decomposed within the liver [24, 25]. Furthermore, MPI provides 3D information over time, while DSA only provides 2D projections over time. An advanced biplane DSA measurements can be post-processed to gain a 3D information over time, which however leads to a doubled radiation exposure. Further, MPI has demonstrated its beneficial usage in several interventional applications such as catheter tracking, stenosis identification and stenosis clearing [26–28]. Catheters and guide wires are coated with magnetic markers to track their position with MPI in real time [29–33]. The first bimodal experiments combining IVOCT and MPI are presented in [34, 35]. The 3D vessel center line can be estimated from static MPI images. With the help of the estimated vessel center line the IVOCT images are oriented in 3D space to reconstruct the vessel volume.

In this work, we track the IVOCT catheter by labeling its tip with an MPI visible marker. Using real time MPI imaging we get the catheter position over time allowing for motion compensation. Both imaging modalities are registered to each other using a time synchronization. An experimental setup, including a custom built IVOCT adapter, enables the generation of different catheter motion profiles. In all experiments the MPI-tracked catheter trajectory is used to reconstruct a 4D IVOCT volume. A straight 3D printed vessel phantom with integrated stenosis is imaged. In a first experiment the plausibility and statistical error of the MPI catheter

tracking is analyzed using a constant catheter velocity. In the following experiments motion artifacts due to catheter bending and heart beat are simulated. The reduction of motion artifacts and their statistical deviation are analyzed. The reduced artifacts in the reconstructed volumes are shown by quantitative measurements using the Dice similarity coefficient (DICE) factor and the estimated stenosis lengths in comparison to the ground-truth shape of the phantom.

## 2 Materials and methods

### 2.1 Experimental setup

The experimental setup is composed of a pre-clinical MPI scanner [36], a custom built IVOCT imaging adapter, a spectral domain OCT system (Telesto I, Thorlabs), and a control unit as shown in Fig 1. A straight vessel phantom with an inner diameter of 2.5 mm and total length of 20 mm is positioned within the MPI field of view (FoV). A stenosis with a length of 1.5 mm and an inner diameter of 1.5 mm is integrated in the phantom (see Fig 1, CAD sketch). A 3D printer (Form 2, Formlabs) based on stereolithography is used to build the phantom out of gray resin. An IVOCT catheter (Dragonfly Duo Kit, Abbott) with an outer diameter of 0.9 mm is used. The catheter consists of an optical fiber covered by a tight and flexible protection, which can rotate freely within a hollow plastic catheter. Within the catheter tip a prism directs the infrared light to the surface. To enable a MPI-based catheter tracking, the catheter tip is coated with a thin layer of magnetic lacquer (1 μL Magneto Magnetic Lacquer, Hand & Nail Harmony), as seen in Fig 1. The lacquer dries quickly on the catheter tip. The OCT images acquired with the marked catheter provide suitable image quality, whereas the phantom structures are still apparent for later segmentation algorithms.

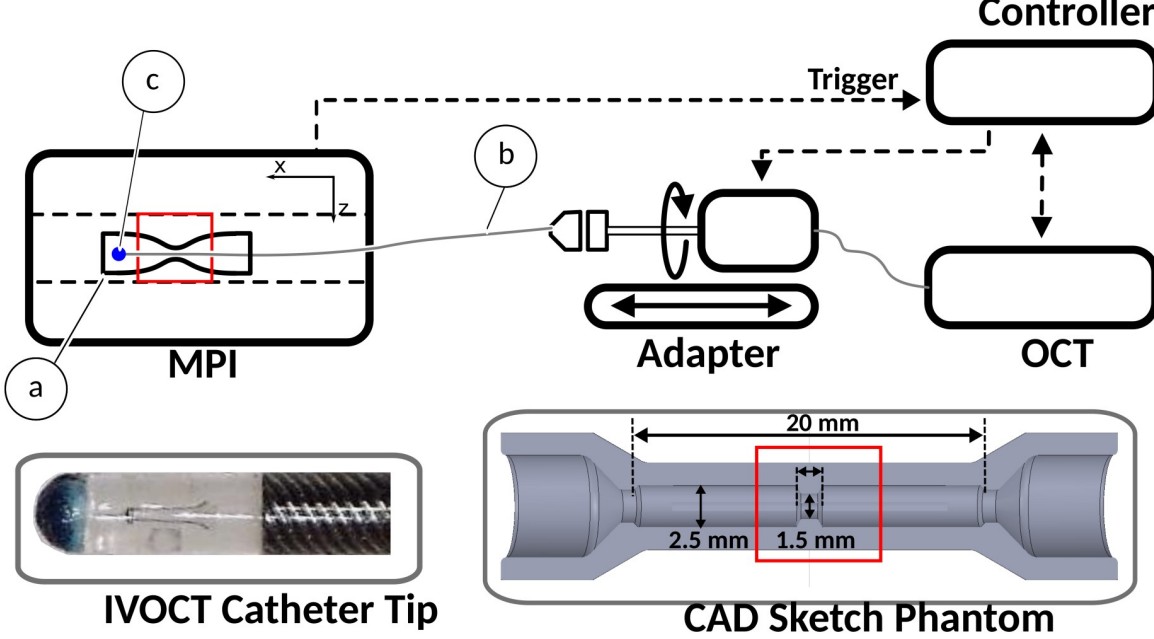

**Fig 1. Experimental setup.** A vessel phantom with a stenosis (a) is positioned within the MPI FoV. In the CAD sketch of the phantom (bottom right), the phantom and entire stenosis dimensions are depicted. The stenosis has a diameter and length of 1.5 mm. Triggered by MPI, an IVOCT catheter (b) is rotated and pulled backwards through the phantom using a custom built adapter. The catheter tip (blue dot) is coated with magnetic lacquer (c) without covering the OCT prism. The cropped MPI FoV is highlighted with a red box.

**2.1.1 MPI acquisition parameters.**   For the MPI measurements a pre-clinical MPI scanner is used together with a custom-built receive coil [37]. The scanner excites the particles with three orthogonal sinusoidal excitation fields with frequencies $f_x$ = 2.5/102 MHz, $f_y$ = 2.5/96 MHz, and $f_z$ = 2.5/99 MHz. The magnetic field strength is set to 12 mT in all three directions while the gradient strength is set to 2.0 T m$^{-1}$ in $z$-direction and 1.0 T m$^{-1}$ in the $x$- and $y$-directions. The imaging period is 21.54 ms which equals a frame rate of 46.43 Hz. The FoV has a size of 24 mm × 24 mm × 12 mm and the MPI data acquisition is conducted with the system software Paravision (Bruker).

In order to reconstruct an MPI image using the frequency space approach [38], a calibration scan is required. This scan moves a small delta sample filled with SPIONs through the FoV while the system response at all attended positions is measured. The acquired data is used to set up the MPI system matrix, which characterizes the relation between the induced voltage signal and the particle distribution. In this work, the system matrix is acquired at 35 × 25 × 13 positions which cover a total volume of 35 mm × 25 mm × 13 mm. To prevent artifacts at the FoV boundaries, the calibration volume is chosen to be larger than the system FoV in all directions [39]. The delta sample has a size of 1 mm × 1 mm × 1 mm and is filled with 1 μL undiluted magnetic lacquer.

**2.1.2 IVOCT acquisition parameters.**   The OCT system with an A-scan rate of $f_{OCT}$ = 91 kHz uses a central wavelength of 1315 nm. The axial OCT FoV is about 2.66 mm in air, whereas each A-scan consists of 512 pixels. The phantom is filled with distilled water yielding a pixel spacing of 4.5 μm between catheter and inner phantom wall, assuming a refractive index of 1.33. The custom-built catheter adapter enables a simultaneous rotation and translation of the IVOCT catheter. A rotational frequency of $f_{rot}$ = 6.25 Hz is used for all experiments. The center pullback velocity $v_0$ = −1.25 mm s$^{-1}$ is varied during the experiments to simulate different motion artifacts. In all experiments, the catheter is pulled back over a total distance of $s$ = 25 mm.

## 2.2 MPI image reconstruction and image processing

In frequency space MPI, the inverse problem to reconstruct an MPI image is treated with a first-order Tikhonov-regularized least-squares approach

$$\underset{c}{\mathrm{argmin}} \quad \|\boldsymbol{S}\boldsymbol{c} - \boldsymbol{u}\|_2^2 + \lambda\|\boldsymbol{c}\|_2^2, \tag{1}$$

where $\boldsymbol{S} \in \mathbb{C}^{M\times N}$ is the MPI system matrix, $\boldsymbol{u} \in \mathbb{C}^M$ is the measurement vector and $\boldsymbol{c} \in \mathbb{R}+^N$ is the particle-concentration vector. This least-squares problem is iteratively solved by using the Kaczmarz method. The Kaczmarz method converges quickly for nearly orthogonal matrices, which is the case for MPI [40, 41]. For the MPI reconstruction and data processing the Julia packages MPIFiles.jl [42] and MPIReco.jl [43] are used. The number of Kaczmarz iterations is set to 3 whereas the regularization parameter $\lambda$ is set to $\lambda = \lambda_0 \cdot 10^{-3}$, where $\lambda_0 = \mathrm{trace}(\boldsymbol{S}^{\mathrm{H}}\boldsymbol{S}) N^{-1}$. These reconstruction parameters have been optimized regarding the visual impression of the reconstructed MPI images.

**2.2.1 MPI-Guided catheter tracking.**   The 4D MPI images are block averaged with a factor of two over time prior to reconstruction, which leads to a temporal resolution of $f_{MPI}$ = 23.2 Hz. The set of MPI images is denoted by $I : \Omega_s \times \mathbb{R} \to \mathbb{R}$ ($\Omega_s \subset \mathbb{R}^3$) with $I(\boldsymbol{x}, t)$ where $\boldsymbol{x}$ is the position and $t$ is the time. The catheter localization is performed in three steps as generally described for more than one marker in [44]. At first, a threshold filter is applied to each image in order to separate the marker from the background. This results in the data set $I^{\mathrm{seg}}$ :

$\Omega_s \times \mathbb{R} \to \mathbb{R}$ with

$$I^{\mathrm{seg}}(\boldsymbol{x}, t) = \begin{cases} 1 & \text{if } I(\boldsymbol{x}, t) \geq \Theta \cdot \max_{\boldsymbol{x}} I(\mathbf{x}, t) \\ 0 & \text{otherwise,} \end{cases} \tag{2}$$

where $\Theta \in [0, 1]$ denotes the relative threshold. In our case the relative threshold is chosen to be $\Theta = 0.35$. In a second step, the connected region $\Omega_1^t \subseteq \Omega_s$, with the highest maximal intensity value $\max I(\Omega_1^t, t)$ is identified by connected-component labeling of $I^{\mathrm{seg}}(\Omega_s, t)$, $t \in \mathbb{R}$. Finally, the position of the catheter marker is obtained by calculating the center of mass

$$\boldsymbol{c}(t) = \frac{\int_{\Omega_1^t} \boldsymbol{x} \cdot I^{seg}(\boldsymbol{x}, t)\mathrm{d}\boldsymbol{x}}{\int_{\Omega_1^t} I^{seg}(\boldsymbol{x}, t)\mathrm{d}\boldsymbol{x}} \tag{3}$$

of the voxel intensities of the corresponding connected region in the MPI image *I*. The accuracy for this sub-voxel approach is within the sub-millimeter range and the catheter position is determined only within cropped MPI FoV robustly. The positions $M_1$ to $M_2$ denote the positions when the catheter enters and leaves this cropped MPI FoV. Hence, we crop the MPI FoV for later 4D reconstruction methods. In *x*-direction the cropped MPI FoV has a length of approximately 10 mm. Outside this cropped MPI FoV the catheter position could not be determined as robust since more image artifacts are introduced by the rotation of the catheter. These outer positions are not considered for the later reconstruction methods. Additionally, outliers are removed with a Ransac algorithm and extreme outliers are excluded via thresholding. In these extreme cases, the images are affected by noise and the localization algorithm falsely detects a high intensity noise voxel as a marker position. Further, the trajectories are smoothed to ensure a continuous trajectory.

## 2.3 Volume reconstruction methods

We refer to two different methods as IVOCT catheter marker tracking (MT) and input parameter (IP) based volume reconstruction, respectively. For both reconstruction methods, the inner phantom wall is segmented in the IVOCT data using a semi-automatic algorithm [34, 35, 45]. Especially in the narrowed phantom parts, some manual corrections are applied. As a result the distance *r* between catheter and phantom wall is given for each A-scan. 3D point clouds are generated based on the MT and IP method, whereas their envelopes are used to quantify the volume reconstructions.

**2.3.1 Input parameter (IP) method.** On the basis of the known input parameters of the custom built adapter (pullback and rotational speed) we take the distances *r* for each OCT A-scans and place a respective point in a 3D coordinate system. We assume a constant pullback and rotational velocity and align the 3D phantom boundary points on a helix with constant pitch $p_0 = v_0/f_{\mathrm{OCT}}$ and angle $\theta_0 = 360° \cdot f_{\mathrm{rot}}/f_{\mathrm{OCT}}$.

**2.3.2 Marker tracking (MT) method.** Using the MPI-guided IVOCT catheter tracking we can arrange the OCT A-scans along the actual catheter trajectory. A temporal synchronization of both imaging systems allows for image registration (Fig 2). The 4D volume reconstruction method is separated in two parts. First, the OCT and MPI data sets are registered via temporal correlation. The measurements are synchronized via a trigger signal sent from MPI to the IVOCT system. The related time events can be seen on the time line in Fig 3. One second after the MPI trigger arises ($t_{\mathrm{trigger}}$), the catheter motion profile and OCT A-scan acquisition starts ($t_{\mathrm{OCT},0}$). The time stamps $t_{M1}$ and $t_{M2}$ are related to the MPI volumes, wherein the

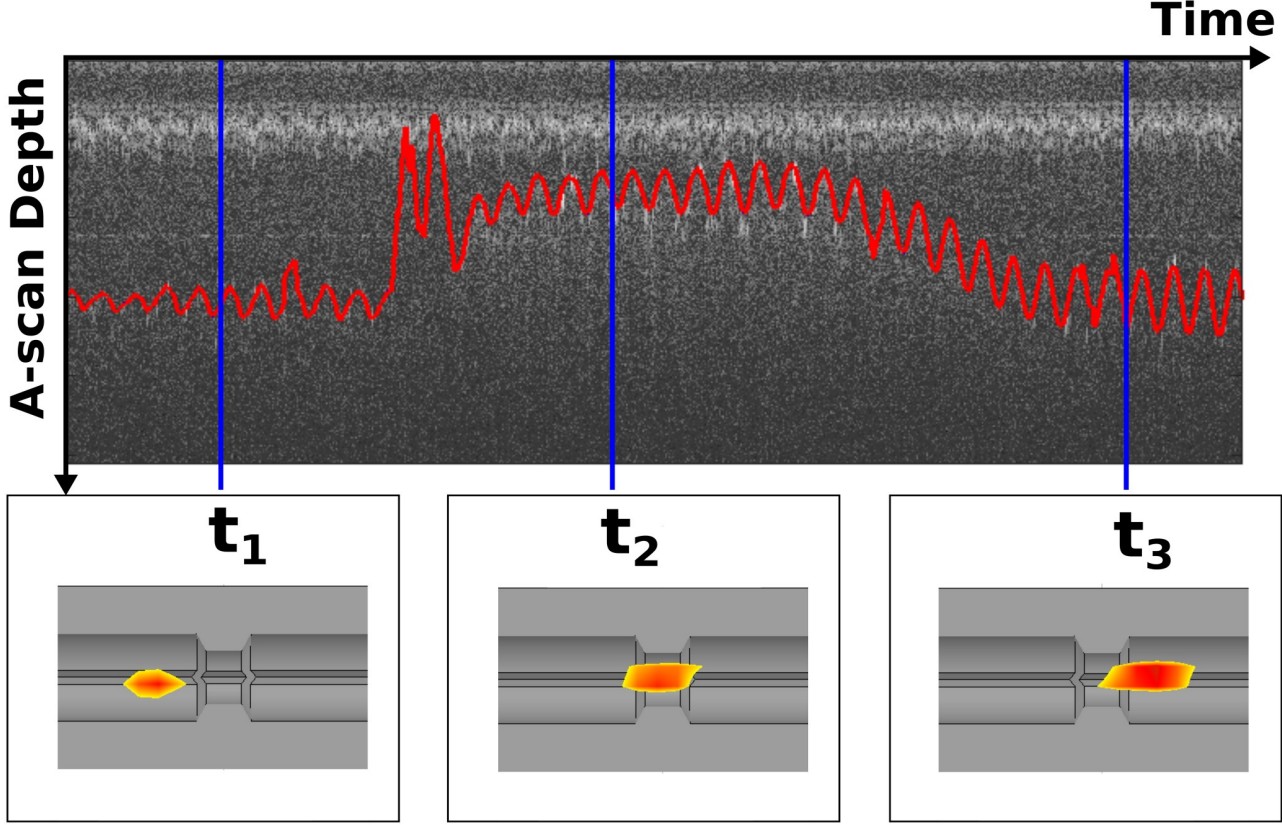

**Fig 2. Exemplary IVOCT and MPI data.** The OCT A-scans are arranged over time (top). The segmented phantom boundary is highlighted in red. For three time stamps $t_i$, the related MPI signals from catheter tip are shown within the CAD sketch (bottom).

catheter tip enters and leaves the cropped MPI FoV. Once the catheter motion profile is finished ($t_{\mathrm{OCT,end}}$) the MPI measurement is stopped subsequently ($t_{\mathrm{MPI,end}}$).

Then, we place points at distance $r$ in 3D space considering both the spatial and temporal dependencies of MPI and OCT data. Due to substantial noise of the $y$- and $z$-component of the estimated 4D catheter trajectory, we only consider the $x$-coordinate (in pullback direction) as catheter position over time. For two successive catheter positions we determine the distance in space $\Delta x$ and time $\Delta t_{\mathrm{MT}}$ and distribute the meanwhile acquired A-scans equidistantly. Based on the given catheter rotation $f_{\mathrm{rot}}$, OCT frequency $f_{\mathrm{OCT}}$, and MPI volume rate $f_{\mathrm{MPI}}$ up to four catheter positions are observed per catheter rotation. Assuming a constant catheter rotation

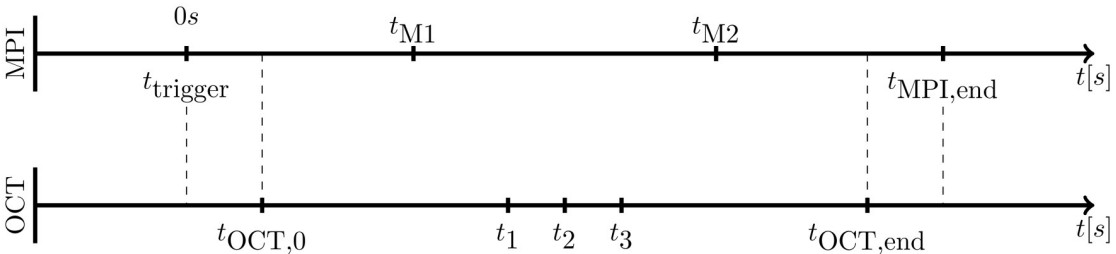

**Fig 3. Time axis for synchronizing OCT device and pullback device with the help of the MPI trigger signal.**

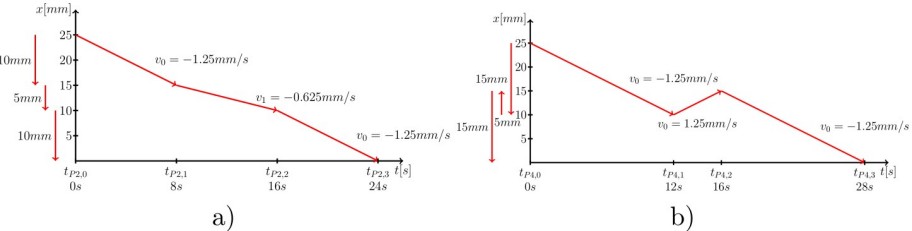

**Fig 4.** a) In case of the BA profile, the catheter is pulled backwards with a velocity $v_0 = -1.25$ mm s$^{-1}$ over the first 10 mm, then the velocity is reduced to $v_1 = -0.625$ mm s$^{-1}$ for the next 5 mm, afterwards the velocity is increased back to $v_0$ for the last 10 mm. b) In case of HBA profile, the IVOCT catheter is pulled backwards over the first 15 mm, then the catheter is moved forward for 5 mm. Last, the catheter is pulled backwards in the original direction again for 15 mm. The catheter moves with the initial velocity $v_0$ for all motion directions.

$f_{\text{rot}}$, the A-scans are oriented with a fixed angle difference $\theta_0$ around the actual catheter trajectory.

## 2.4 Experiments

We perform three experiments with the stenosis phantom repeating each experiment three times. As a first experiment, we conduct a standard pullback profile (SP) with a constant pull-back velocity $v_0$ and a pullback distance *s*.

As a second experiment, a bending artifact profile (BA) is used to simulate the non-linear pullback of the catheter when the catheter is decelerated due to a bending and is then suddenly accelerated due to its elastic material. At first, the catheter is pulled back with velocity $v_0$ for the first 10 mm. Afterwards the catheter is simulated to be stuck and its velocity is set to $v_1 = -0.625$ mm s$^{-1}$ for the next 5 mm. Finally, the velocity is set back to the initial velocity $v_0$ for the last 10 mm to simulate the elastic contraction of the catheter. The distances over time of the BA profile are shown in Fig 4a).

As a third experiment, we perform a measurement with a heart beat motion artifact (HBA) profile. A heart beat artifact is related to the heart contraction and the relative vessel motion w. r.t. the IVOCT catheter. This artifact results in multiple acquisitions of the same blood vessel part due to a back and forth movement of the vessel (Fig 5). We use a catheter motion profile (HBA) that simulates this relative motion. The velocity is set to $v_0$ for the first 15 mm.

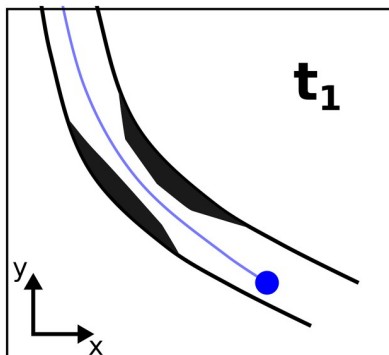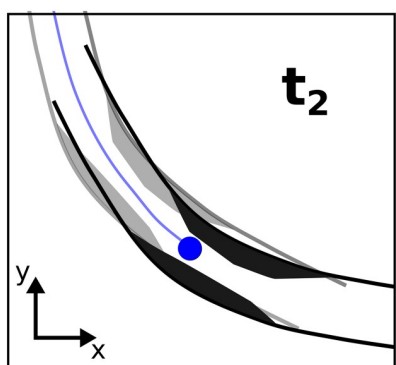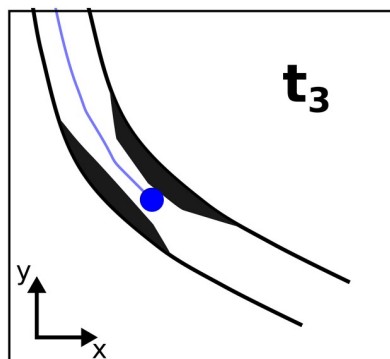

**Fig 5. Exemplary sketch of heart beat motion artifact.** Due to heart contraction the imaged artery is deformed for time stamp $t_2$. Meanwhile, the catheter tip (blue dot) moves continuously backwards. After heart contraction ($t_3$) the artery gets back to its original shape ($t_1$). Again, the catheter motion is continued in between. This relative motion between catheter and artery leads to multiple IVOCT imaging of the sketched stenosis (black).

Subsequently, the velocity is inverted to $-v_0$ for the following 5 mm to imitate the heart beat movement. Afterwards the velocity is adjusted back to $v_0$ for the last 15 mm. The distance over time of the HBA profile is shown in Fig 4b).

The described phantom experiments are performed *in-vitro* and do not involve human subjects. The datasets are acquired with the described imaging devices (pre-clinical MPI scanner, spectral-domain OCT system). The MPI data processing is implemented in Julia, while the OCT data processing is written in Matlab.

## 3 Results

The results are divided into three parts. First, the positions and the resulting velocities determined by 4D MPI catheter tracking are validated for three motion profiles SP, BA, and HBA. The mean absolute error (MAE) is calculated for the distance traveled only in $x$ and for the distance traveled in $x$, $y$, $z$. The same is done for the velocities of the profiles. Second, we compare the IP and the MT volume reconstruction using the IVOCT and MPI data from the standard profile. The influence on both reconstruction methods in terms of bending artifacts is analyzed for the BA profile. Additionally, the HBA profile is used to investigate heart beat artifacts on both reconstruction methods. Third, the DICE factor is calculated for both reconstruction methods. In addition, the stenosis length is quantified for all reconstruction methods/profiles and compared to its ground-truth value.

### 3.1 Statistical validation of 4D MPI catheter tracking

For the standard profile the distance in $x$ over time between $M_1 = 18$ mm and $M_2 = 6$ mm is shown in Fig 6a). From 18 mm to 11 mm the tracked $x$ positions (black) are in good agreement with the expected $x$ positions (red).

Between 11 mm to 6 mm the tracked $x$ positions (black) seem to diverge slightly from the expected values (red). The mean values are used to fit a regression line (blue). The MAE for the distance in $x$ is 0.44 mm ± 0.44 mm. The absolute error (AE) of the velocity using the regression line is 0.21 mm s$^{-1}$ with a relative error (RE) of 16.8% as given in Table 1. For the BA profile the distance in $x$ over time between 18 mm and 6 mm is presented in Fig 6b). Overall the tracked $x$ positions (black) are in good agreement with the expected $x$ positions (red). Only in the first segment a small deviation is visible. Again, all three measurements are shown

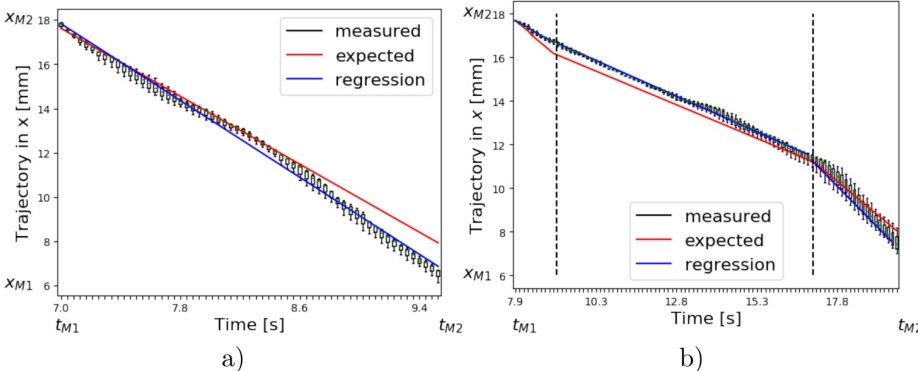

**Fig 6.** a) MPI measurements for standard profile: The measured distance in $x$ over time between 18 mm and 11 mm is in good agreement with the expected values. Expect in the last part the positions in $x$ marginally deviate. b) MPI measurements for BA profile: The first change of velocity is not captured by the measurement. After the first change the measured distances in $x$ over time are in agreement with the different velocity $v_0$ and $v_1$ with only slight deviations.

**Table 1. The mean absolute error (MAE) is given for the distance in *x*-direction with its standard deviation (SD).** Additionally, the absolute error (AE) along with the relative error (RE) of the velocity using the 3D regression line between $t_{M1}$ and $t_{M2}$ is also reported.

| Errors | Standard profile |
|---|---|
| MAE Trajectory 1D-*x* [mm] | 0.44 ± 0.44 |
| AE (RE) Velocity [mm/s] | 0.21 (16.8%) |

as a box plot and illustrate the distribution of the tracked positions. The mean values are used to determine a regression line (blue).

The MAE for the distance in *x* is 0.26 mm ± 0.16 mm for the first segment, 0.35 mm ± 0.11 mm for the second segment and 0.20 mm ± 0.22 mm for the third segment. The AE and RE regarding the velocity of the regression in all three segments for the BA profile are given in Table 2.

For the HBA profile the distances in *x* over time between 18 mm and 6 mm are shown in Fig 7a. Overall, the tracked *x* positions resemble the movement of the catheter being pulled back and forth. The turning points between velocities $-v_0$, $v_0$ and again $-v_0$ can be clearly identified. However, in the first two segments the velocity is underestimated as the tracked *x* positions are not in full accordance with the expected *x* positions (red). In the third segments the tracked *x* positions agree with the expected values. The distances in *y* and *z* over time are presented in Fig 7b and 7c) and shows that the stenosis phantom has been inserted slightly diagonal as the *y*-values increase and the *z*-values decrease depending on the *x*-position. For a straight insertion we would expect a straight line in both dimensions. Only at the time points when the velocities change the tracked positions in *x* differ from the expected positions in *x*. The three measurements are depicted as box plots to show the distribution of the measurements. The regression lines for each segment are plotted in blue. The mean absolute error for the distance in *x* is 0.64 mm ± 36 mm for the first segment, 0.51 mm ± 55 mm for the second segment and 0.38 mm ± 45 mm for the third segment. In Fig 7d) the velocity in *x* over time is shown and the inversion of the velocity is visible. The absolute and relative error of the velocity using the regression line in *x* are 0.38 mm/s (30.0%) for the first segment, 0.49 mm/s (39.4%) for the second segment and 0.04 mm/s (3.2%) for the third segment. The errors regrading the HBA profile are given in Table 3.

## 3.2 Volume reconstructions

In Fig 8 the 4D volume reconstructions are compared for all motion profiles and both reconstruction methods. The volumes are shown with *x* cropped to the MPI FoV. A ground-truth volume with boundary information created by the parameters from the CAD sketch is depicted as a reference. The 4D boundary points are colored related to the underlying time, whereas the color map is shifted with respect to the time values of the positions $x_{M_1} = 18mm$.

**Table 2. The mean absolute error (MAE) for the distance in *x*-direction for the BA profile is given with its standard deviation (SD).** The AE along with the RE of the velocity using the 3D regression line between $t_{M1}$ and $t_{M2}$ is also reported.

| Errors | BA profile segment 1 | BA profile segment 2 | BA profile segment 3 |
|---|---|---|---|
| MAE Trajectory 1D-*x* [mm] | 0.26 ± 0.16 | 0.35 ± 0.11 | 0.20 ± 0.22 |
| AE (RE) Velocity [mm/s] | 0.44 (35.4%) | 0.07 (10.8%) | 0.22 (17.9%) |

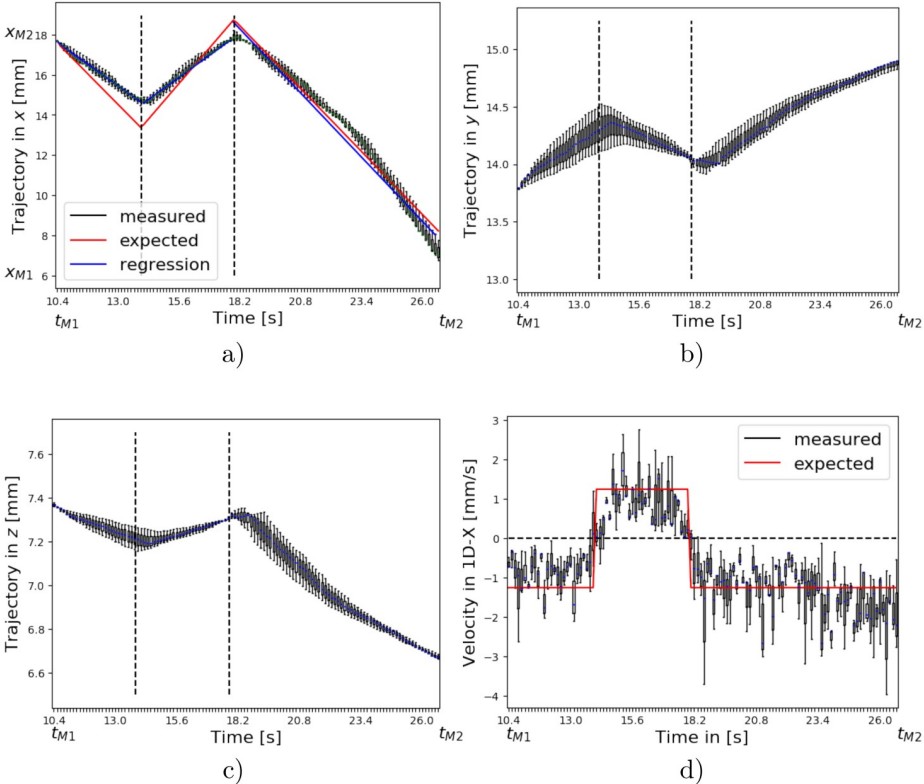

**Fig 7.** a) The measured distances in *x* agree with the set catheter movement. The turning points are clearly visible. The velocity, however, is underestimated in the first two segments. In the third segment the measured distances in *x* agree with the expected positions in *x*. b) The measure distance in *y* and c) *z* shows that the stenosis phantom is inserted slightly diagonal and the back and forth movement is also noticeable in the *y* and *z* dimension. d) The inversion of the velocity is visible and the mean velocity value are within the range of the expected velocities. However, the spread of the velocity is quite high.

The envelopes of all volume reconstructions show deviations compared to the ground-truth volume. The stenosis lengths are highlighted with red arrows. The MT reconstruction method leads to stenosis lengths and relative positions that are almost equal to the ground-truth volume for all motion profiles. The IP reconstruction method shows a larger deviation of the stenosis relative position. Furthermore, an obvious deviation of the depicted stenosis length using the IP volume reconstruction method are depicted for the BA and HBA profiles. Especially, for the BA profile with underlying deceleration of the catheter, the length is obviously increased.

In order to consider the complete pullback time for the BA and HBA profile, the related 4D volumes without cropping the *x*-axis are shown in Figs 9 and 10, respectively. Considering the BA profile (Fig 9), the overall IP volume results in an increased length with constant helical pitch $p_0$. In contrast, the MT volume does not overestimate the total volume and especially the

**Table 3. The mean absolute error for the distance in *x*-direction with its standard deviation (SD) is presented for the HBA profile.** The absolute and relative error of the velocity using the 3D regression line between $t_{M1}$ and $t_{M2}$ is also reported.

| Errors | HBA profile segment 1 | HBA profile segment 2 | HBA profile segment 3 |
|---|---|---|---|
| MAE Trajectory 1D-*x* [mm] | 0.64 ± 0.36 | 0.51 ± 0.55 | 0.38 ± 0.45 |
| AE (RE) Velocity [mm/s] | 0.38 (30.0%) | 0.49 (39.4%) | 0.04 (3.2%) |

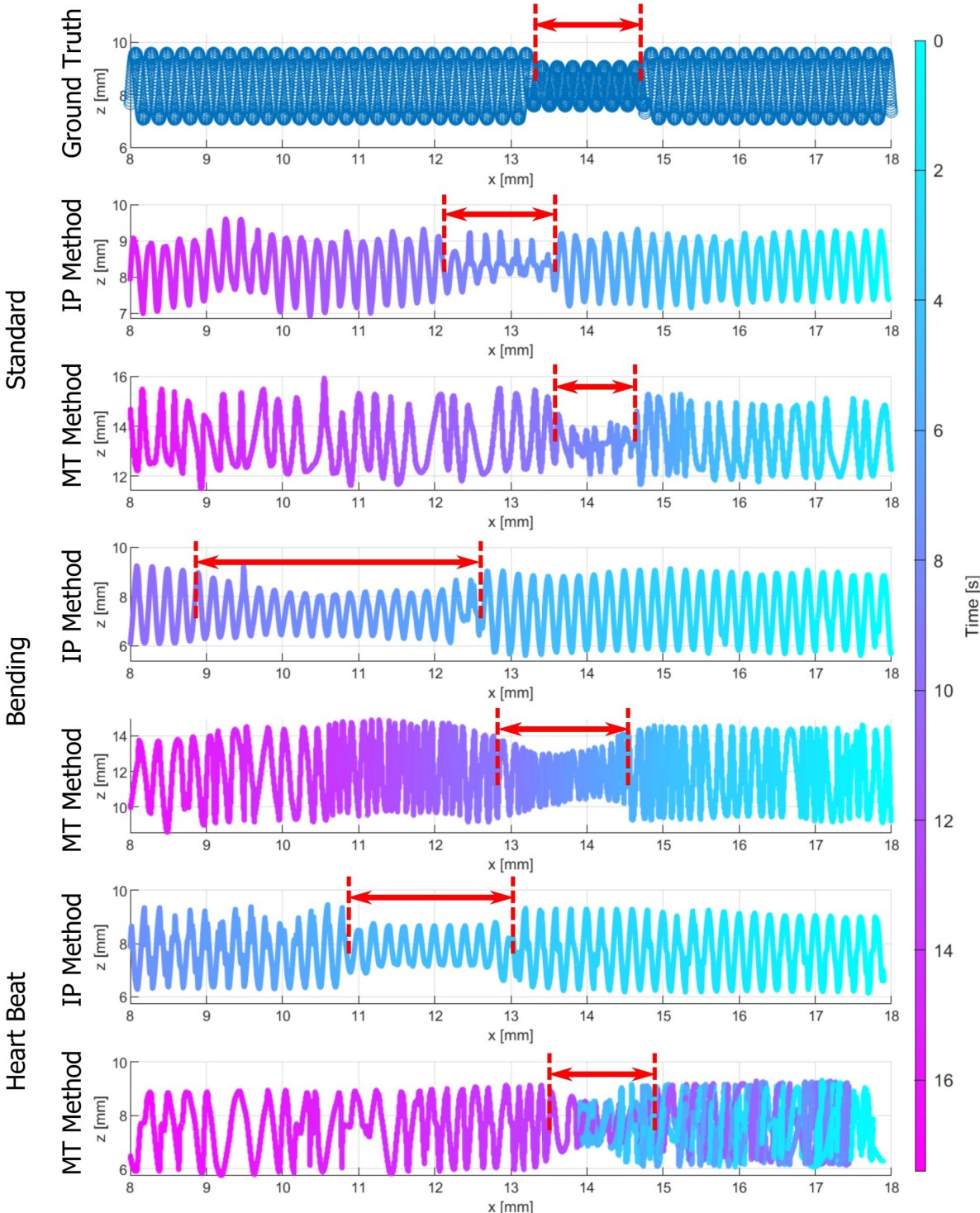

**Fig 8. Reconstructed volumes for all motion profiles w.r.t. the ground-truth volume (top) for the cropped MPI FoV.** The distances $x = 18$ and $x = 8$ mm correspond to the time points $t_{M1}$ and $t_{M2}$. The IP and MT volume reconstructions are labeled (left). The phantom boundary points are colored w.r.t. the time color map (right). The stenosis lengths are depicted with red arrows.

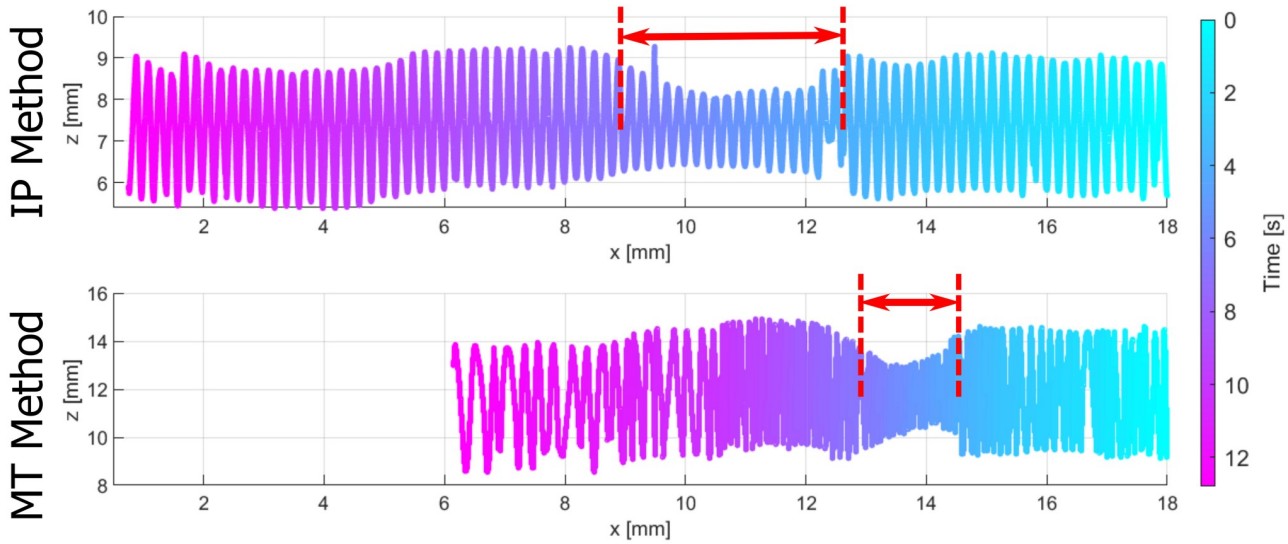

**Fig 9. Complete IP volume reconstruction compared to MT volume reconstruction for the bending profile.**

stenosis length. The varying catheter velocity, as depicted in Fig 6b), is apparent for the MT method by different densities of boundary points between $x_{M_1}$ and $x_2 = 10.2$ mm compared to the points between $x_2$ and $x_3 = 8$ mm. In case of the HBA profile (Fig 10), the IP volume also shows a relevant overestimation of the total volume. Furthermore, in the volume reconstruction beyond $x = 8$ mm a second stenosis appears. The MT volume again presents an improved reconstruction method. Considering the tracked catheter motion, the 3D boundary points are arranged over time such that several boundary points overlay each other between $x_1$ and $x_4 = 14$ mm. Hence, the colored 4D volume (bottom) represents a catheter trajectory with a turning point within the stenosis.

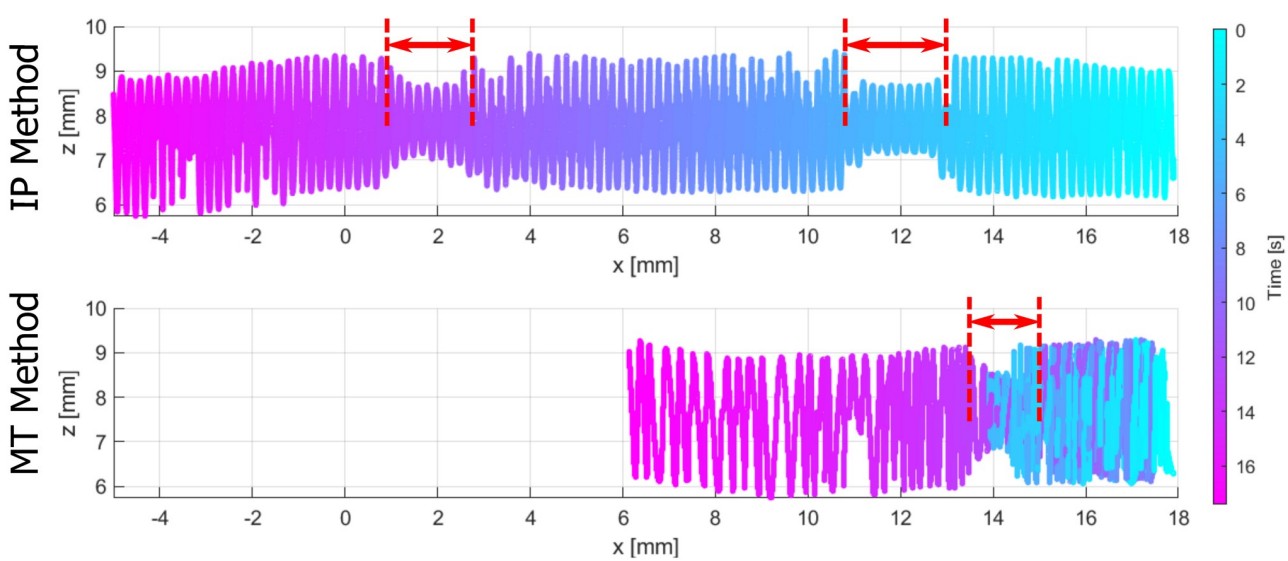

**Fig 10. Complete IP volume reconstruction compared to MT volume reconstruction for the heart beat profile.**

**Table 4. DICE quantification and related stenosis lengths in mm for 3D and 4D reconstruction methods for motion profiles SP, BA, and HBA, respectively.**

|  | SP profile | BA profile | HBA profile |
|---|---|---|---|
| DICE IP | 0.86 | 0.66 | 0.58 |
| DICE MT | 0.88 | 0.89 | 0.86 |
| Stenosis IP (RE) | 1.52 (1.3%) | 3.91 (160%) | 3.82 (154%) |
| Stenosis MT (RE) | 1.15 (23%) | 1.49 (0.6%) | 1.19 (21%) |

## 3.3 Quantitative volume results

We determine the envelopes of the 3D boundary points of the IP and MT methods and quantify the volume reconstruction results using the DICE metric

$$DICE = \frac{1}{N}\sum_{i=1}^{N} 2 \cdot \frac{|U_i \cap V_i|}{|U_i| + |V_i|},$$ (4)

whereas $U_i$ are the 2D projected shapes of the reconstructed envelopes for method IP and MT, respectively, compared to the ground-truth 2D projected shapes $V_i$ for all angles from 1 to $N = 180°$. The mean DICE for all repetitions are listed in Table 4. The stenosis length is determined in $x$-direction as full at half width of the envelope decay of the volume shapes for all reconstructions, profiles and experimental repetitions. In case of the HBA profile, the stenosis length is determined as summation of the two stenosis lengths.

## Discussion

The 4D catheter trajectory is tracked by the MPI for three different catheter motion profiles. The statistical validation of all motion profiles and repetitions reveal small MAEs in $x$-direction of around 0.5mm, which is in good accordance with the estimated determination accuracy [44]. The ground truth for the trajectory in terms of the position in $x$, $y$ and $z$ directions is not known, since it is hardly possible to track the catheter's position within the MPI scanner with a second instance, e.g., an optical system. The ground truth for the trajectory is only known in terms of the pullback velocity and distance in 3D over a defined period. Therefore, the calculated MAEs in $x$-direction contain a small uncertainty because the $y$ and $z$ ground truth positions are assumed to be constant zero. The absolute and relative errors of the velocities determined by a regression line in 3D are comparable to the ground truth velocity of the IVOCT adapter. They show varying relative errors in the range of 3.2%–39% for all motion profiles and their segments. For the BA profile the first change of velocity could not be captured by the measured values. One reason could be that the velocity change is close to the border of the FoV. The voxel intensities representing the marked catheter tip have circle shaped form in the MPI images. If this circle shape has not entered the FoV completely, the center of mass localization algorithm might misinterpret the position. In addition, in case of the HBA profile deviations occur around the turning points seen in Fig 7a), which lead to an underestimation of the velocity. These deviations can be linked to the catheter setup with a proximal actuator such that the pullback is increased by the shrinkage and stretching of the flexible catheter. It is also worth noting that the back and forth motion of the catheter is also visible in the $y$- and $z$-positions seen in Fig 7b and 7c) as the vessel phantom is not placed perfectly in accordance with the $x$-axis.

The novel MT volume reconstruction method based on MPI catheter tracking demonstrates a qualitative improvement in comparison to the IP method (Fig 8). Even in case of the SP profile without additional motion artifacts, the IP method shows worse results by means of

the DICE metric. The illustrated results of the motion artifact profiles underline the need of a catheter tracking over time. In addition to the catheter tracking in 3D [7, 10, 34, 35], the time synchronization of the IVOCT and MPI data leads to an optimized arrangement of OCT A-scans in 3D space. The DICE metrics and stenosis lengths in Table 4 emphasize the relevant errors in case of the IP method. Nevertheless, inaccuracies in volume shapes occur for all methods and profiles (DICE< 0.9), as other imaging artifacts have an influence on the IVOCT and MPI data. For example, non-uniform rotational distortions (NURD) of the catheter might appear due to the catheter setup. Furthermore, the boundary segmentation in the IVOCT data as well as the catheter tip segmentation in the MPI data contain inaccuracies.

In future work, the results can be further improved by a correction of additional artifacts and image enhancements. On the one hand, a rotation tracking with MPI may be possible with an asymmetric marking of the catheter tip. On the other hand, the phantom centerline and catheter trajectory can be tracked using a multi-contrast MPI imaging approach [29–32, 46, 47] visualizing the marker and the blood pool tracer inside the phantom. Both approaches can be used to minimize the effect of NURD artifacts.

## Conclusion

A novel approach for MPI-guided IVOCT catheter tracking is presented considering both the 3D catheter trajectory and the time synchronization of IVOCT and MPI data in order to reconstruct volumes of a known vessel phantom shape in 4D. A DICE coefficient of up to 89% is achieved for different IVOCT motion artifact studies. The presented approach estimates the stenosis length for simulated artifacts more precisely with a relative error of up to 0.6% in comparison to 160% of the standard method.

## Author Contributions

**Conceptualization:** Florian Griese, Sarah Latus, Matthias Schlüter, Matthias Graeser, Matthias Lutz, Tobias Knopp.

**Data curation:** Florian Griese, Sarah Latus.

**Formal analysis:** Sarah Latus.

**Funding acquisition:** Alexander Schlaefer, Tobias Knopp.

**Methodology:** Florian Griese, Sarah Latus, Tobias Knopp.

**Software:** Florian Griese, Sarah Latus, Tobias Knopp.

**Validation:** Florian Griese, Sarah Latus.

**Visualization:** Florian Griese, Sarah Latus.

**Writing – original draft:** Florian Griese, Sarah Latus.

**Writing – review & editing:** Florian Griese, Sarah Latus, Matthias Schlüter, Matthias Graeser, Alexander Schlaefer, Tobias Knopp.

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
