## [Decision Letter · Decision Letter 0]

3 Jan 2020

PONE-D-19-33061

In-Vitro MPI-Guided IVOCT Catheter Tracking in Real Time for Motion Artifact Compensation

PLOS ONE

Dear Mr. Griese,

Thank you for submitting your manuscript to PLOS ONE. After careful consideration, we feel that it has merit but does not fully meet PLOS ONE’s publication criteria as it currently stands. Therefore, we invite you to submit a revised version of the manuscript that addresses the points raised during the review process.

We would appreciate receiving your revised manuscript by Feb 17 2020 11:59PM. To enhance the reproducibility of your results, we recommend that if applicable you deposit your laboratory protocols in protocols.io, where a protocol can be assigned its own identifier (DOI) such that it can be cited independently in the future. For instructions see: http://journals.plos.org/plosone/s/submission-guidelines#loc-laboratory-protocols

We look forward to receiving your revised manuscript.

Kind regards,

Wolfgang Rudolf Bauer, M.D., Ph.D.

Academic Editor

PLOS ONE

Journal Requirements:

2. In your methods section, please provide more information about the provenance of the dataset, and clarify whether the study involved human subjects.

Reviewers' comments:

Reviewer's Responses to Questions

**Comments to the Author**

1. Is the manuscript technically sound, and do the data support the conclusions?

Reviewer #1: Yes

2. Has the statistical analysis been performed appropriately and rigorously? 

Reviewer #1: Yes

3. Have the authors made all data underlying the findings in their manuscript fully available?

Reviewer #1: Yes

4. Is the manuscript presented in an intelligible fashion and written in standard English?

Reviewer #1: Yes

5. Review Comments to the Author

Reviewer #1: The manuscript "In-Vitro MPI-Guided IVOCT Catheter Tracking in Real Time for Motion Artifact Compensation" describes a combination of IntraVascular Optical Coherence Tomography with Magnetic Particle Imaging. The authors use MPI for determining the position as well as the velocity of the catheter during OCT. The use of a magnetic lacquer enables MPI to detect the marked areas of the catheter as has been previously shown in literature. This approach allows a radiation-free bimodal application of significant (pre-)clinical relevance. Today's standard uses as reference DSA (X-ray) imaging methods.

The manuscript is well written and understandable, the figures are comprehensive. Nevertheless, there are a few issues that need to be resolved prior to a possible publication:

l22: Please choose: "proximally rotated catheters" or "rotated proximal catherters" depending on what your want to express.

l74: You are claiming the lacquer does not affect the OCT beam profile. Please either give a suitable reference or show your measurements supporting this claim.

l92: then -> than

l97: Either "contains" or "consists of" but not "contains of".

l136: What is your criterium for declaring a value an "outlier"?

l137: What kind of smoothing did you apply?

l184: While mathematically correct, please avoid speaking of "decreased" in this context, rather use "set". A decrease leading to a higher absolute velocity if counter-intuitive for the reader.

fig4+6: I recommend to combine these two figures only displaying the contents of 4b and 6a. A constant velocity (4a) needs no depiction and I do not see the benefit of having the velocities graphically presented in 6b.

l210: You are stating that the tracked positions are in good agreement with expected values from 18mm to 10mm. Looking at the error bars of your measured values I would rather cut this argument short at 11mm.

fig7 (caption): Here you state the good agreement was from 18mm down to 6mm. This is (a) in contradiction to your own statement in the text (18mm-10mm) and also please consider my previous comment.

l215: The values of AE and RE given here do not agree with the ones given in tab1. Please verify!

l218: You are addressing "a small deviation" of the measured values from the expected ones. Looking at fig7b the measurements seems not to capture the first change in velocity at all since it starts at a wrong slope. Please discuss this result in more detail! Are there issues with higher velocities?

l227 and fig8: Similarly to the above case the velocity seems to be underestimated while you are constating a good accordance. Please comment on the too small slopes for higher velocities.

l236/7: Three times the units (mm) are missing.

l239/40 and tab3: The values given for AE/RE do not agree between table and text.

Good job!

Your Reviewer

6. PLOS authors have the option to publish the peer review history of their article (what does this mean?). If published, this will include your full peer review and any attached files.

Reviewer #1: No

---

## [Author Response · Author response to Decision Letter 0]

20 Jan 2020

The responds to the reviewer and editor comments is given in die document "Answers to the reviewers".

---

## [Decision Letter · Decision Letter 1]

10 Mar 2020

In-Vitro MPI-Guided IVOCT Catheter Tracking in Real Time for Motion Artifact Compensation

PONE-D-19-33061R1

Dear Dr. Griese,

We are pleased to inform you that your manuscript has been judged scientifically suitable for publication and will be formally accepted for publication once it complies with all outstanding technical requirements.

With kind regards,

Wolfgang Rudolf Bauer, M.D., Ph.D.

Academic Editor

PLOS ONE

Additional Editor Comments (optional):

Reviewers' comments:

Reviewer's Responses to Questions

**Comments to the Author**

1. If the authors have adequately addressed your comments raised in a previous round of review and you feel that this manuscript is now acceptable for publication, you may indicate that here to bypass the “Comments to the Author” section, enter your conflict of interest statement in the “Confidential to Editor” section, and submit your "Accept" recommendation.

Reviewer #1: All comments have been addressed

2. Is the manuscript technically sound, and do the data support the conclusions?

Reviewer #1: Yes

3. Has the statistical analysis been performed appropriately and rigorously? 

Reviewer #1: Yes

4. Have the authors made all data underlying the findings in their manuscript fully available?

Reviewer #1: Yes

5. Is the manuscript presented in an intelligible fashion and written in standard English?

Reviewer #1: Yes

6. Review Comments to the Author

Reviewer #1: All my remarks to the original manuscript have been fully addressed. From my side there are no more issues to be resolved prior to a possible publication.

Your reviewer.

7. PLOS authors have the option to publish the peer review history of their article (what does this mean?). If published, this will include your full peer review and any attached files.

Reviewer #1: Yes: Volker C. Behr

---

## [Editor Report · Acceptance letter]

13 Mar 2020

PONE-D-19-33061R1 

*In-Vitro* MPI-Guided IVOCT Catheter Tracking in Real Time for Motion Artifact Compensation 

Dear Dr. Griese:

I am pleased to inform you that your manuscript has been deemed suitable for publication in PLOS ONE. Congratulations! Your manuscript is now with our production department. 

With kind regards,

on behalf of

Prof. Wolfgang Rudolf Bauer 

Academic Editor

PLOS ONE